# Private sector delivery of quality care for maternal, newborn and child health in low-income and middle-income countries: a mixed-methods systematic review protocol

Samantha R Lattof  , Blerta Maliqi

Department of Maternal, Newborn, Child, Adolescent Health and Ageing, World Health Organization, Geneva, Switzerland

**Correspondence to**
Dr Samantha R Lattof;
lattofs@who.int

## ABSTRACT

**Introduction** To accelerate progress to reach the sustainable development goals for ending preventable maternal, newborn and child deaths, it is critical that both the public and private health service delivery systems invest in increasing coverage of interventions to sustainably deliver quality care for mothers, newborns and children at scale. Although various approaches have been successful in high-income countries, little is known about how to effectively engage and sustain private sector involvement in delivering quality care in low-income and middle-income countries. Our systematic review will examine private sector implementation of quality care for maternal, newborn and child health (MNCH) and the impact of this care. This protocol details our intended methodological and analytical approaches, based on the Preferred Reporting Items for Systematic Reviews and Meta-Analyses (PRISMA) reporting guideline for protocols.

**Methods and analysis** Following the PRISMA approach, this systematic review will include quantitative, qualitative and mixed-methods studies addressing the provision of quality MNCH care by private sector providers. Eight databases (Cumulative Index to Nursing and Allied Health, EconLit, Excerpta Medica Database, International Bibliography of the Social Sciences, Popline, PubMed, ScienceDirect, Web of Science) and two websites will be searched for relevant studies published between 1 January 1995 and 30 June 2019. For inclusion, studies in low-income and middle-income countries must examine at least one of the following critical outcomes: maternal morbidity or mortality, newborn morbidity or mortality, child morbidity or mortality, quality of care, experience of care and service utilisation. Depending on the data, analyses could include meta-analysis, descriptive quantitative statistics, narrative synthesis and thematic synthesis. Quality will be assessed using tools for qualitative and quantitative studies.

**Ethics and dissemination** Formal ethical approval is not required for this research, as the secondary data are not identifiable. Findings from this review will be used to develop models for effective collaboration of the private and public sectors in implementing quality of care for MNCH. In addition to publishing our findings in a peer-reviewed journal, the findings will be shared through the

## STRENGTHS AND LIMITATIONS OF THIS STUDY

⇒ This mixed-methods systematic review protocol is the first to examine private sector delivery of quality maternal, newborn and child healthcare in low-income and middle-income countries.
⇒ By using the Preferred Reporting Items for Systematic Reviews and Meta-Analyses checklist, we increase the potential usefulness, clarity and transparency with which we will report the results from this study.
⇒ To minimise potential bias from the publication of positive results and to increase the validity of this systematic review, we will make a concerted effort to supplement our electronic database searches with grey literature and programmatic reports.
⇒ We acknowledge a risk of bias in locating studies, as it is possible evidence on the provision of quality healthcare by the private sector may be disseminated internally beyond public reach.
⇒ For transparency and quality, the protocol includes a codebook and data extraction template.

Quality of Care Network, relevant mailing lists, webinars and social media.
**PROSPERO registration number** CRD42019143383

## INTRODUCTION

The private sector plays a key role in delivering sexual and reproductive health services. It provides a substantial proportion of family planning services among women aged 15–49 years in Asia (45%), Latin America and the Caribbean (44%), and sub-Saharan Africa (28%).[1] In certain countries, like Democratic Republic of Congo and Nigeria, the private sector's family planning market share exceeds 60%.[2] One in five births in low-income and middle-income countries occurred with care delivered by the private sector.[3] Despite the private sector's expanding role in many countries, the quality of services varies. To

accelerate progress to reach the sustainable development goals for ending preventable maternal, newborn and child deaths, it is critical that both the public and private health service delivery systems invest in increasing coverage of interventions to sustainably deliver quality care at scale.

Among countries in the Network for Improving Quality of Care for Maternal, Newborn and Child Health (the Network), the private sector addresses an increasing volume of maternal, newborn and child health (MNCH) care needs. The Network, a consortium of 11 countries and their technical partners, aims to halve maternal and newborn deaths and stillbirths in health facilities in 5 years' time.[4] While the Network's efforts to achieve this ambitious goal have largely focused on strengthening the public health sector, members of the Network recognise that private providers (ie, non-government providers, for-profit businesses) are an important source of healthcare and have a role to play in improving quality care. However, little is known about how to effectively engage and sustain private sector involvement in delivering quality care in low-income and middle-income countries. This gap must be addressed, if the Network is to achieve its aims of reducing maternal and newborn deaths and stillbirths.

The engagement and contribution of the private sector in implementing quality care standards, developing and identifying best practices for delivering quality MNCH care, and strengthening health systems for delivering quality are areas of great potential that require immediate attention. There is a need to understand what can be done to create, nurture and encourage a vibrant private sector that is fully engaged in improving and sustaining quality of care for mothers, newborns and children. We aim to begin filling these knowledge gaps by conducting a systematic review that addresses the following research questions:

### Primary research questions

1. How and to what extent does the provision of quality healthcare by the private sector affect morbidity and mortality among mothers, newborns and children?
2. How and to what extent does provision of quality healthcare by the private sector affect utilisation of services by mothers, newborns and children?
3. How effective and efficient is the private sector at delivering quality of care?
4. Among mothers, newborns and children using healthcare provided by the private sector, what are their experiences of care?

### Secondary research questions

5. What mechanisms exist for engaging the private sector in delivering quality MNCH services? These mechanisms may allow for demonstrating accountability for quality MNCH services; developing, implementing and sharing MNCH standards and quality improvement approaches, developing and sharing quality of care imple-

mentation packages for MNCH, and developing and sharing policies and plans for quality MNCH. Private sector engagement, as recognised by WHO, includes three broad categories: (1) including private actors in the development of public health policy and the development of ownership and contracting arrangements; (2) influencing private sector behaviour through regulatory and financing policy tools; and (3) assigning 'private attributes' to public sector organisations (eg, by giving them managerial autonomy and exposing them to market forces and incentives).[5]
6. What types of regulatory and collaborative service delivery models exist between the public and private sectors to deliver quality of care? These models may include the private sector's involvement in national MNCH quality of care structures and coordinating mechanisms.

This protocol details our intended methodological and analytical approaches for the systematic review, based on the Preferred Reporting Items for Systematic Reviews and Meta-Analyses (PRISMA) reporting guideline for protocols.[6]

## METHODS AND ANALYSIS

This systematic review will use the PRISMA checklist for reporting systematic reviews and meta-analyses so as to ensure clarity and transparency.[7]

### Inclusion/exclusion criteria

This review focuses on studies examining the private sector's involvement in delivering quality healthcare to pregnant people, mothers, newborns and children (aged nine years and under) in low-income and middle-income countries. Using the World Bank Atlas method for the current 2020 fiscal year, this review will include countries classified as having low-income economies, lower-middle-income economies and upper-middle-economies.[8] The population, interventions, control, outcomes, timeframe, setting (PICOTS) criteria, detailed in table 1, provide an overview of our inclusion criteria. Quantitative, qualitative and mixed-methods studies will be considered, provided that they report on at least one of the following critical outcomes:

► Maternal morbidity or mortality.
► Newborn morbidity or mortality.
► Child morbidity or mortality.
► Components of quality of care, defined by WHO as 'the extent to which health care services provided to individuals and patient populations improve desired health outcomes. In order to achieve this, health care must be safe, effective, timely, efficient, equitable and people-centred'. (p1046)[9]
► Experience of care, including respectful care.
► Service utilisation.

Journal articles and grey literature (eg, reports, book chapters), both peer-reviewed and non-peer-reviewed, will be eligible for inclusion. In recognition of the rapid

**Table 1** The population, interventions, control, outcomes, timeframe, setting (PICOTS) criteria used in the systematic review

| PICOTS | |
|---|---|
| Populations | Pregnant people, mothers, newborns and children (aged nine years and under). |
| Interventions | Delivery of quality maternal, newborn and/or child health services by the private sector. |
| Control | Not necessary. |
| Outcomes | Quantitative, qualitative or mixed-methods data on:<br>► Maternal morbidity or mortality.<br>► Newborn morbidity or mortality.<br>► Child morbidity or mortality.<br>► Components of quality of care (ie, safety, effectiveness, timeliness, efficiency, equity, people-centred care).<br>► Experience of care, including respectful care.<br>► Service utilisation. |
| Timeframe | 1 January 1995 to 30 June 2019. |
| Setting | Low-income and middle-income countries. |

increase in public-private collaborations for health during the late 1990s,[10] items must be published from 1 January 1995 to 30 June 2019. Items will be eligible if published in English, French, German or Italian. For inclusion, quality must be a specific focus among the interventions being implemented, such as quality assessment, quality improvement, clinical quality, perceived quality or the delivery of quality care. Interventions addressing universal health coverage will also be considered as focusing on quality, since quality health services are a fundamental feature of universal health coverage.[11]

As we are focused on service delivery, we are limiting the private sector to providers who deliver direct medical care (eg, private health facilities, private health providers, civil society organisations delivering care, charities delivering care). Thus, private sector entities that do not deliver direct medical care will be excluded. For example, we will exclude private organisations offering social service support to orphaned children, private pharmaceutical providers (including pharmacies) and private health insurance companies.

We will exclude studies reporting on aggregated service delivery data (ie, public sector and private sector outcome data combined), as we are only interested in outcomes resulting from service delivery by the private sector. Studies addressing allied fields like family planning, malaria and HIV/AIDS will be excluded unless they include specific MNCH interventions (eg, post-partum family planning, integration of family planning with maternal health services, abortion services, treatment of malaria in children, prevention of mother-to-child transmission of HIV) as a specific focus or aim. If commentaries, reviews and editorials do not introduce new evidence, then we will exclude them in favour of the original study or studies being discussed. Study protocols will also be excluded.

## Search strategy and terms

We will search the following eight electronic databases that were assessed for likely coverage, availability and relevance of literature on the private sector:

► Cumulative Index to Nursing and Allied Health.
► EconLit.
► Excerpta Medica Database.
► International Bibliography of the Social Sciences.
► Popline.
► PubMed.
► ScienceDirect.
► Web of Science.

While we considered the LILACS (Latin American and Caribbean Health Sciences Literature) database, our test search returned no results. A test search in Open Grey returned one extraneous result, so we excluded that database as well. Based on discussions with experts and an initial assessment of the electronic databases, we suspect that a significant body of literature on the private sector is published as grey literature or programmatic reports. Thus, we will make a concerted effort to supplement our electronic database searches with expert-recommended articles and reports that we will obtain by sending a standardised request for literature to a list of experts and working groups identified in collaboration with colleagues at WHO.

Given our interest in locating grey literature, we will also conduct a targeted search of publications on two websites: Health Care Provider Performance Review (HCPPR) and the Maternal healthcare markets Evaluation Team (MET) at the London School of Hygiene & Tropical Medicine. HCPPR is a database of over 700 studies from a systematic review on the effectiveness of strategies to improve healthcare provider performance in low-income and middle-income countries.[12] HCPPR's search strategy included extensive searching of document inventories for unpublished studies. MET conducts multidisciplinary research on the role of public and private health sectors in delivering maternal healthcare. Its website includes both peer-reviewed literature and grey literature. We acknowledge risk of bias in locating studies. Our efforts to locate evidence on the provision of quality healthcare by the private sector may be limited, specifically when such evidence is only disseminated internally where it is beyond public reach.

Our searches will be conducted using combinations of search terms detailed in table 2. These terms aim to capture all studies examining the private sector, quality of care and MNCH. The terms were developed and tested for sensitivity, and they will be adapted to the basic search particulars (eg, wildcards (*), capacity for complex searches) of each electronic database.

## Screening process

Searches and application of the inclusion/exclusion criteria will be conducted according to the PRISMA flow

**Table 2** Search terms and their combinations

| (1) Private sector | (2) Quality of care | (3) MNCH |
|---|---|---|
| private sector | quality | matern* |
| for-profit | | pregnan* |
| for profit | | mother* |
| public-private | | newborn* |
| private enterprise* | | infant* |
| NGO | | child* |
| non-government* | | pediatric* |
| | | paediatric* |
| | | neonat* |

MNCH, maternal, newborn and child health; NGO, non-governmental organisation(s).

approach. Citation abstracts for all items uncovered by our searches will be exported into EndNote for screening. After first removing duplicates, remaining items will be screened for inclusion on the basis of title and abstract (TIAB). If inclusion/exclusion cannot be determined based on the TIAB, then the item will be pushed forward for full-text screening. SRL will conduct all TIAB and full-text screening, seeking guidance from BM on items considered borderline or problematic.

### Data extraction

The data extraction form (online supplementary annex A) will be piloted with five randomly selected studies. Any necessary changes will be discussed among the authors prior to full data extraction. Based on the guidance in our codebook (online supplementary annex B), SRL will extract data for all studies into the data extraction template in Excel. Data will be extracted on the following categories:

► Background information (eg, author, date, setting, study objective).
► Intervention background information (eg, implementing agency, geographic level, study population).
► Intervention details (eg, intervention recipients, nature of intervention, dimensions of quality care).
► Critical outcomes (both quantitative and qualitative):
    – Maternal morbidity or mortality.
    – Newborn morbidity or mortality.
    – Child morbidity or mortality.
    – Quality of care.
    – Experience of care, including respectful care.
    – Service utilisation and efficiency.
► Secondary outcome (both quantitative and qualitative):
    – Infant and/or child growth.
► Evaluation/study details (eg, study type, data type, intervention claims, strategy effectiveness, cost data).
► Study quality (qualitative and quantitative).

Study quality and risks of bias will be assessed in duplicate using quality assessment tools for quantitative and qualitative studies. For quantitative studies, we will use the Effective Public Health Practice Project's (EPHPP) quality assessment tool that rates public health studies on selection bias, study design, confounders, blinding, data collection methods, withdrawals and dropouts, intervention integrity and analysis.[13] The accompanying reviewer's dictionary provides details on how to evaluate each section (ie, strong, moderate or weak) with the process resulting in 'excellent' agreement between reviewers on the study's final grade.[13 14] For qualitative studies, we will use Miltenburg *et al*'s[15] quality assessment tool based on criteria developed by Walsh and Downe.[16] This tool rates qualitative studies' scope and purpose, design, sampling strategy, analysis, interpretation, reflexivity, ethical dimensions and relevance and transferability.[16] Like the EPHPP tool, Miltenburg *et al*'s tool involves rating each section as strong, moderate or weak and then using these ratings to determine the paper's overall quality rating (ie, strong, moderate or weak).

### Data synthesis

Once the data have been extracted, we will analyse the quantitative and qualitative findings separately. Depending on the quantitative studies and data available in the final inventory, a meta-analysis may be appropriate to examine the outcomes of private sector interventions on the provision of quality MNCH care. Data extracted on intervention details (see online supplementary annexes A and B, variables 'Nature of Intervention$1' through 'Nature of Intervention$6') will be reviewed for their suitability in meta-analyses. We will examine the effect of each type of private sector intervention affecting the delivery of care (eg, introduction or change to on-site support for quality improvement, introduction or change to data systems) on our six critical outcomes. When appropriate, we will conduct meta-analyses and generate forest plots in RevMan V.5.3.5 in non-Cochrane mode.[17] For interventions in which the study sample size is large and interstudy heterogeneity is minimal, we will use fixed model effect; otherwise, we will use random effect size.[18] The meta-analysis will be limited to randomised control trials and observational studies rated by the EPHPP tool as strong or moderate quality, and meta-biases of included studies will be assessed using ROBIS, a tool for assessing risk of bias in systematic reviews.[19] If high heterogeneity exists between study designs, interventions, populations and definitions, then we will instead present the quantitative findings using a narrative synthesis to report the data thematically with tables of descriptive quantitative statistics and study outcomes.

For the qualitative data, we will enter the verbatim findings into NVivo for a three-step thematic synthesis.[20] Each line of text will initially be coded based on its content and meaning. All codes will be included in a codebook with new codes added when appropriate. The codes will then be related together to develop categories and linkages between the data. After reviewing the data for analytical themes, we will make inferences about the provision of

MHCH care by the private sector. We suspect that this type of synthesis will be particularly well suited to data on patients' experiences of care by the private sector and existing mechanisms for engaging the private sector in delivering quality MNCH services.

## Patient and public involvement

The design of this systematic review protocol did not involve patients, though we hope the final inventory of studies included in this review will report on patients' experiences of care. The need for this systematic review was initiated by discussions with public sector health officials in Network countries.

## FINAL SEARCH STRATEGY BY DATABASE

The full electronic search strategies for all databases, including limits and filters used, appear below.

## Cumulative Index to Nursing and Allied Health

Search strategy: We will search all sets of search terms (table 2).

### Search options

► Search mode: Boolean/phrase.
► Limit results:
  – Published date: January 1995 to June 2019.
  – Language: English, French, German, Italian.
  Search terms: ("private sector" OR for-profit OR "for profit" OR public-private OR "private enterprise*" OR NGO OR non-government*) AND (quality) AND (matern* OR pregnan* OR mother* OR newborn* OR infant* OR child* OR pediatric* OR paediatric* OR neonat*)

## EconLit

Search strategy: We will search all sets of search terms (table 2).

### Search options

► Search mode: Boolean/phrase.
► Limit results:
  – Published date: January 1995 to June 2019.
  Search terms: ("private sector" OR for-profit OR "for profit" OR public-private OR "private enterprise*" OR NGO OR non-government*) AND (quality) AND (matern* OR pregnan* OR mother* OR newborn* OR infant* OR child* OR pediatric* OR paediatric* OR neonat*)

## Excerpta Medica Database

Search strategy: We will search all search terms (table 2) using the multi-field search in the abstract field.

### Search options

► Limit results:
  – Publication year: 1995–current.
  – Language: English, French, German, Italian.

Search terms: (private sector OR for-profit OR for profit OR public-private OR private enterprise* OR NGO OR non-government*) AND (quality) AND (matern* OR pregnan* OR mother* OR newborn* OR infant* OR child* OR pediatric* OR paediatric* OR neonat*)

## Health Care Provider Performance Review

Search strategy: For this systematic review database on the effectiveness of strategies to improve healthcare provider performance in low-income and middle-income countries, we will obtain a list of all HCPPR's included grey literature directly from the project investigator. Grey literature on maternal, newborn and/or child health services will be included for review.

### Search options

► Limit results:
  – Publication year: 1995–2019.

## International Bibliography of the Social Sciences

Search strategy: We will search all sets of search terms (table 2) using the advanced search feature. To minimise extraneous results returned during test searches, searches will be conducted 'anywhere except full text'. Thus, the searches will use title and abstract, in line with our initial plans for TIAB screening.

### Search options

► Limit results:
  – Publication date: 1 January 1995–30 June 2019.
  – Language: English, French, German, Italian.
  Search terms: noft(("private sector" OR for-profit OR "for profit" OR public-private OR "private enterprise*" OR NGO OR non-government*) AND (quality) AND (matern* OR pregnan* OR mother* OR newborn* OR infant* OR child* OR pediatric* OR paediatric* OR neonat*))

## Maternal healthcare markets Evaluation Team

Search strategy: On the MET publications page, we will export all documents that have not been published in peer-reviewed journals (eg, reports, policy briefs). Peer-reviewed publications appear in journals that have been indexed in other electronic databases included in this systematic review.

## Popline

Search strategy: We searched a modified set of search terms (table 2). Since Popline was retired on 1 September 2019, we relied on an earlier search that did not include the search term "neonat*". The database does not permit users to include wildcards inside quotation marks, so we split "private enterprise*" into two terms: "private enterprise" and "private enterprises."

### Search options

► Filters:
  – Year: Published since 1995.
  – Language: English, French, German, Italian.

Search terms: (("private sector" OR for-profit OR "for profit" OR public-private OR "private enterprise" OR "private enterprises" OR NGO OR non-government*) AND (quality) AND (matern* OR pregnan* OR mother* OR newborn* OR infant* OR child* OR pediatric* OR paediatric*))

## PubMed

Search strategy: We will search all sets of search terms (table 2) using the advanced search builder.

### Search options
► Limit results:
  – Publication dates: From 1 January 1995 to 30 June 2019.
  – Language: English, French, German, Italian.

Search terms: ("private sector" OR for-profit OR "for profit" OR public-private OR "private enterprise*" OR NGO OR non-government*) AND (quality) AND (matern* OR pregnan* OR mother* OR newborn* OR infant* OR child* OR pediatric* OR paediatric* OR neonat*)

## ScienceDirect

Search strategy: Since this database does not support wildcards (*) or more than eight Boolean connectors per field, we will search a modified set of search terms (table 2) using the advanced search feature and limiting keywords to those most widely used. We will split the terms between articles and title, abstract or keywords, as detailed below:

### Search options
► Limit results:
  – Year(s): 1995–2019.

### Search terms
► Find articles with these terms: (maternal OR maternity OR newborn OR child OR children OR childhood OR childbirth OR pregnancy OR pregnant)
► Title, abstract or keywords: (quality) AND ("private sector" OR for-profit OR "for profit" OR public-private OR "private enterprise" OR "private enterprises" OR NGO OR non-governmental)

## Web of Science

Search strategy: We will search all sets of search terms (table 2) using the advanced search feature and topic field tag.

### Search options
► Limit results:
  – Year(s): 1995–2019.
  – Language: English, French, German, Italian.

Search terms: ("private sector" OR for-profit OR "for profit" OR public-private OR "private enterprise*" OR NGO OR non-government*) AND (quality) AND (matern* OR pregnan* OR mother* OR newborn* OR infant* OR child* OR pediatric* OR paediatric* OR neonat*)

## ETHICS AND DISSEMINATION

Formal ethical approval is not required for this research, as the publicly available secondary data are not identifiable.

This systematic review, which we anticipate concluding by 1 June 2020, will provide evidence on different models that can be drawn upon by countries in planning and implementing their national plans and processes for quality of care. Findings from this review will be used to develop models for effective collaboration of the private and public sectors in implementing quality of care for MNCH. In addition to publishing our findings in a peer-reviewed journal, the findings will be shared through the Quality of Care Network, relevant mailing lists, webinars and social media.

Should we need to amend this protocol following its publication, we will ensure that future publications arising from this protocol provide the date of each amendment, describe the change(s) and report the rationale for the change(s).

**Acknowledgements** We wish to thank David Clarke (WHO), Joby George (Save the Children), and Wilson Were (WHO) for their contributions to the wider discussions around this research; Moïse Muzigaba (WHO) for his comments on an earlier draft of this manuscript; and Alex Rowe (Centers for Disease Control and Prevention) for sharing information on the HCPPR database, including a list of grey literature references.

**Contributors** BM and SRL conceived the idea for the review. SRL designed and wrote the first draft of the protocol. Both authors contributed to subsequent revisions and approved the protocol prior to its submission. SRL is the guarantor.

**Funding** This work was supported by MSD for Mothers and the Maternal, Newborn, Child, Adolescent Health and Ageing Department of the World Health Organization. MSD for Mothers had no role in the design and development of the study protocol or the decision to publish.

**Competing interests** None declared.

**Patient consent for publication** Not required.

**Provenance and peer review** Not commissioned; externally peer reviewed.

**ORCID iD**
Samantha R Lattof http://orcid.org/0000-0003-0934-1488

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
