## [Reviewer comments · BMJ Open]

ARTICLE DETAILS

TITLE (PROVISIONAL)	Private sector delivery of quality care for maternal, newborn, and child health in low- and middle-income countries: A mixed-methods systematic review protocol
AUTHORS	Lattof, Samantha R.; Maliqi, Blerta

VERSION 1 – REVIEW

REVIEWER	VICTOR MOGRE University for Development Studies, School of Medicine and Health Sciences, Ghana
REVIEW RETURNED	29-Aug-2019

GENERAL COMMENTS	In this this protocol, authors intend to systematically review the literature on the contribution of the private sector towards improving the delivery maternal and child health in low-and middle-income countries. The protocol is well written and if well executed will contribute to our understanding of how the private sector contribute to maternal and child health delivery in LMICs. However, I have a few concerns that requires the attention of the authors. Authors should include low-and middle-income countries in the title of the protocol. E.g. "Private sector delivery of quality care for maternal, newborn, and child health in low-and middle-income countries: A mixed-methods systematic review protocol" Authors should define Low-and middle-income countries that they will be using in the review. Authors should provide more information on the quality assessment tool they will be using. What is the name of the tool; what aspects does it measure; scoring, etc. Authors should include a discussion section in which they describe how the findings significance of the findings and how they will be disseminated.
---

REVIEWER	Dr. Ranadip Chowdhury Society For Applied Studies India
REVIEW RETURNED	06-Sep-2019

GENERAL COMMENTS	This is a very relevant meta-analysis in the context of maternal, newborn and child health. I have some comments: 1. One of the primary objectives of the meta-analysis is "How effective and efficient is the private sector at delivering the quality
--

	of care?" How this objective will be analyzed? What will be the comparator group? 2. Will infant or children growth be captured as one of the outcomes? 3. Neonate should be included as one of the search terms. 4. A brief plan of analysis section is required for the quantitative data where meta-analysis will be possible. 5. A brief discussion section is required.
--	---

VERSION 1 – AUTHOR RESPONSE

Reviewer: 1

Reviewer Name

VICTOR MOGRE

Institution and Country

University for Development Studies, School of Medicine and Health Sciences,
Ghana

Please state any competing interests or state 'None declared':

None

Please leave your comments for the authors below

In this this protocol, authors intend to systematically review the literature on the contribution of the private sector towards improving the delivery maternal and child health in low-and middle-income countries. The protocol is well written and if well executed will contribute to our understanding of how the private sector contribute to maternal and child health delivery in LMICs.

Authors' reply: Thank you.

However, I have a few concerns that requires the attention of the authors.

Authors should include low-and middle-income countries in the title of the protocol. E.g.

"Private sector delivery of quality care for maternal, newborn, and child health in lowand middle-income countries: A mixed-methods systematic review protocol" **Authors' reply:** We have amended the protocol's title to note that the review focuses on low- and middle-income countries.

Authors should define Low-and middle-income countries that they will be using in the review.

Authors' reply: Under our inclusion/exclusion criteria, we have clarified that we are using the World Bank Atlas method for the current 2020 fiscal year to determine low- and middle-income countries. As such, our review will include countries classified by the World Bank as

having low-income economies, lower-middle-income economies, and upper-middle-economies. This detail also appears in our codebook (Annex B).

Authors should provide more information on the quality assessment tool they will be using. What is the name of the tool; what aspects does it measure; scoring, etc. **Authors' reply:** We have amended our 'Data extraction' section to include additional details of the quality assessment tools. Additionally, these tools are available in our codebook (Annex B) and data extraction template (Annex A), where we note what we are measuring and how we score each item.

Authors should include a discussion section in which they describe how the findings significance of the findings and how they will be disseminated.

Authors' reply: Since *BMJ Open* considers the addition of a discussion section for a protocol manuscript as discretionary, we have not included one at this time. Once we complete our systematic review and are in a position to identify our findings of significance, we will make sure to highlight them in the discussion section of our ensuing manuscript. For now, details of our dissemination plan for any findings of significance appear under 'Ethics and dissemination.'

Reviewer: 2

Reviewer Name

Dr. Ranadip Chowdhury

Institution and Country

Society For Applied Studies

India

Please state any competing interests or state 'None declared':

None

Please leave your comments for the authors below

This is a very relevant meta-analysis in the context of maternal, newborn and child health. I have some comments:

1. One of the primary objectives of the meta-analysis is "How effective and efficient is the private sector at delivering the quality of care?" How this objective will be analyzed? What will be the comparator group?

Authors' reply: Efficacy and efficiency, two of the six dimensions of quality care, now appear in our definition of 'quality care' and are captured in our systematic review's data extraction tool (Annex A). Our tool captures which dimensions of quality care each study seeks to

address within the private health sector (see ‘Reference to Quality\$4’ in Annexes A and B). Specific outcome data on quality of care, which by definition includes effective (i.e., “providing services based on scientific knowledge and evidence-based guidelines,” p. 1046) and efficient (i.e., “delivering health care in a manner which maximizes resource use and avoids waste,” p. 1046) health care (Tuncalp 2015), will be extracted in our data extraction tool under ‘Critical Outcome\$7A’ through ‘Critical Outcome\$7C’. Quantitative and/or qualitative data examining the efficacy and efficiency of the private sector would then be analysed as described under ‘Data synthesis.’ Based on our PICOTS for this systematic review, comparator groups are not a prerequisite for inclusion, though we would certainly report if studies examining these dimensions of quality care included comparator groups. Initial test searches and screenings suggest that much data on these particular dimensions of quality care may need to be analysed qualitatively.

2. Will infant or children growth be captured as one of the outcomes? **Authors’ reply:** We have added infant and child growth as a secondary outcome in our ‘Data extraction’ section.

3. Neonate should be included as one of the search terms.
Authors’ reply: We have added ‘neonat*’ as a search term in order to capture all possible variations (e.g., neonatal, neonate, neonates).

4. A brief plan of analysis section is required for the quantitative data where metaanalysis will be possible.
Authors’ reply: Under ‘Data synthesis,’ we have expanded upon our plan of analysis for quantitative data where meta-analysis will be possible.

5. A brief discussion section is required.
Authors’ reply: Since *BMJ Open* considers the addition of a discussion section for a protocol manuscript as discretionary, we have not included one at this time.

VERSION 2 – REVIEW

REVIEWER	Dr. Ranadip Chowdhury Society For Applied Studies India
REVIEW RETURNED	12-Oct-2019
GENERAL COMMENTS	The authors have addressed all comments adequately.